# Influence of Process Parameter and Alloy Composition on Misoriented Eutectics in Single-Crystal Nickel-Based Superalloys

**DOI:** 10.3390/ma16124477

**Published:** 2023-06-20

**Authors:** Tobias Wittenzellner, Shieren Sumarli, Zijin Dai, Ocson Cocen, Helge Schaar, Fu Wang, Dexin Ma, Andreas Bührig-Polaczek

**Affiliations:** 1Foundry Institute, RWTH Aachen University, 52072 Aachen, Germany; dai-@outlook.de (Z.D.); h.schaar@gi.rwth-aachen.de (H.S.); d.ma@gi.rwth-aachen.de (D.M.); sekretariat@gi.rwth-aachen.de (A.B.-P.); 2Laboratory of Neutron Scattering and Imaging, Paul Scherrer Institute, Forschungsstrasse 111, 5232 Villigen, Switzerland; shieren.sumarli@psi.ch; 3Haute École Arc Conservation-Restauration, HES-SO University of Applied Sciences and Arts Western Switzerland, 2000 Neuchâtel, Switzerland; ocson.cocen@he-arc.ch; 4State Key Laboratory for Manufacturing System Engineering, School of Mechanical Engineering, Xi’an Jiaotong University, Xi’an 710049, China; fuwang@xjtu.edu.cn

**Keywords:** nickel-based superalloy, micro-structure, directional solidification, nucleation, crystal orientation, carbide

## Abstract

The nucleation and the growth of misoriented micro-structure components in single crystals depend on various process parameters and alloy compositions. Therefore, in this study, the influence of different cooling rates on carbon-free, as well as carbon-containing, nickel-based superalloys was investigated. Castings were carried out using the Bridgman and Bridgman–Stockbarger techniques under industrial and laboratory conditions, respectively, to analyze the impact of temperature gradients and withdrawing rates on six alloy compositions. Here, it was confirmed that eutectics could assume a random crystallographic orientation due to homogeneous nucleation in the residual melt. In carbon-containing alloys, eutectics also nucleated at low surface-to-volume ratio carbides due to the accumulation of eutectic-forming elements around the carbide. This mechanism occurred in alloys with high carbon contents and at low cooling rates. Furthermore, micro-stray grains were formed by the closure of residual melt in Chinese-script-shaped carbides. If the carbide structure was open in the growth direction, they could expand into the interdendritic region. Eutectics additionally nucleated on these micro-stray grains and consequently had a different crystallographic orientation compared with the single crystal. In conclusion, this study revealed the process parameters that induced the formation of misoriented micro-structures, which prevented the formation of these solidification defects by optimizing the cooling rate and the alloy composition.

## 1. Introduction

Dendritic solidification occurs during the solidification process of nickel-based superalloys under industrial process parameters. During dendrite formation, carbides nucleate into the interdendritic regions either homogeneously from the residual melt or heterogeneously on particles such as ceramic particles from the mold [1,2]. Consequently, they form a random crystallographic orientation [2]. Carbide morphology depends on the carbon content, as well as the cooling conditions. They can acquire morphologies ranging from blocky and acicular to Chinese-script-shaped [1,3,4,5,6,7].

γ/γ′-eutectic is the last phase to solidify from the residual melt; its solidification mechanisms have been controversially discussed. Some studies describe the formation of the coarse γ′-phase of the γ/γ′-structure as a result of a peritectic transformation ($L_{4}+\;\gamma \to \;\gamma^{\prime }$) [8,9,10]. At the interface between the residual melt and the γ-dendrites, γ′ nucleates from the primary γ-phase, since its lattice mismatch is small. Subsequently, the γ′-phase grows into the melt and into the γ-dendrites. The growth into the melt is faster than into the matrix due to a higher diffusion rate in the liquid than in the solid phase. After the completion of the peritectic reaction, growth of the γ′-phase into the residual melt follows. In the end, fine lamellar γ/γ′ is formed by epitaxial growth ($L_{5}\to \;\gamma +\gamma^{\prime }$) [8,9,10].

In contrast, other studies suggest that a eutectic reaction leads to the formation of the interdendritic γ/γ′-structure [11,12,13]. At the beginning of this reaction, the melt ahead of the solidification front of the γ-dendrites is enriched in γ′-forming elements, such as Al, Ta, and Ti, and has significant undercooling. Next, a certain region in the solid–liquid interface grows faster during incubation time and advances into the liquid region. A fine lamellar γ/γ′-structure is subsequently formed by the eutectic reaction. The latent heat is released in this process, resulting in the reduction of undercooling and delaying the eutectic reaction [14,15]. Consequently, the γ′-phase has sufficient time to absorb more γ′-forming elements and to release the γ-forming elements, leading to coarser γ′-phases in thicker γ-channels around the fine structure of the core [13].

Wang et al. demonstrated in various studies that eutectics can also nucleate on the carbides [16,17]. As a result, the eutectics adopt the crystallographic orientation of the carbides and thus deviate from the single crystal [17]. Thereby, a semi-coherent interface, which is always shifted on the eutectic side, is formed [16]. Eutectic nucleation on carbides is possible, despite the different lattice parameters due to undercooling that occurs around the carbides where non-carbide-forming elements, such as Al, Co, Cr, and Ni, accumulate [16]. After nucleation, Wang et al. described a co-growth mechanism between carbide and eutectic, as the carbide releases the eutectic-forming elements and the eutectic releases the carbide-forming ones [16].

Differently oriented γ/γ′-eutectics have also been reported, which are assumed to nucleate homogeneously from the melt [17]. However, they have only been studied in carbon-containing alloys, thus not completely eliminating the possibility of nucleation on carbides that are just not visible in the observed 2D cross-section. In consequence, these misoriented eutectics appear as micro-stray grains in the interdendritic region, which can lead to recrystallization or growth of defects during heat treatment, causing deterioration of the high-temperature properties of the single-crystal parts [17]. In addition, these micro-stray grains are reported to result in crack initiation [16], illustrating the need to eliminate misoriented micro-defects in single crystals.

In this study, the hypothesis of the homogeneous formation of γ/γ′-eutectic was investigated using carbon-free alloys. In order to understand the nucleation mechanism of eutectics on carbides, the alloy composition and the process parameters, i.e., withdrawal rate and temperature gradient, were varied. It was concluded how this effect could be influenced or prevented.

## 2. Materials and Methods

In order to verify the defect formation under industrial conditions, the first set of experiments was performed using the VIM IC 5S Bridgman furnace from ALD Vacuum Technologies GmbH (Hanau, Germany). For this purpose, investment casting shells with a chill plate diameter of 250 mm were produced. They consisted of four cylindrical specimens with a diameter of 16 mm and four blade geometries each (Figure 1a). (The blade geometry is not part of this paper). The grain selector method was used to produce a single crystal derived from six different nickel-based superalloy compositions (Table 1). Thermocouples were placed in the specimens to measure the cooling rate. Directional solidification was carried out at withdrawal rates of 1, 2.5, 4, and 5.5 mm/min.

Up to three single-crystal cylindrical samples produced from the industrial Bridgman furnace (withdrawal rate 4 mm/min) were partially remelted and directionally solidified in the Bridgman–Stockbarger furnace from Linn High Therm GmbH (Hirschbach, Germany). The single-crystal structure was reproduced using the seeding technique, whose exact method has already been described in [18]. This process enabled investigations at higher temperature gradients and lower withdrawal rates compared with the industrial system. To determine the solidification conditions, thermocouples were installed in these specimens through eroded holes in the center of the specimen. The withdrawal rates of 0.25, 1, and 4 mm/min were applied in this set of experiments. An overview of the process parameters, namely withdrawal rate and temperature gradient, together with the resulting cooling conditions of both experimental sets, are summarized in Figure 1b and Table 2. The temperature gradient was measured between 1673 and 1473 K. The cooling rate was calculated with the following formula: cooling rate = temperature gradient ∗ withdrawal rate. In this study, the Bridgman experiments were named “W + withdrawal rate” (for example “W1”); the Bridgman–Stockbarger experiments were labeled “Q-W + withdrawal rate” (for example “Q-W1”). All combinations of alloys and withdrawal rates are listed in Table 3. 

Metallographic specimens were prepared from both longitudinal and transverse sections of the single crystals in an as-caste state to investigate the micro-structural components and the misoriented structures. The Optical microscope (OM) Axio from Zeiss (Oberkochen, Germany) was used to observe the micro-structure of the specimens etched with waterless Kalling reagent (80 mL EtOH, 40 mL HCl und 2 g ${\mathrm{CuCl}}_{2}$). Using these micrographs, the primary dendrite arm spacing (PDAS) was investigated. Deep etching for up to 20 min with aqua regia (20 mL HNO und 60 mL HCl) enabled the three-dimensional (3D) morphology of the carbides to be captured by the scanning electron microscope (SEM) Zeiss Supra 55 VP (Oberkochen, Germany). The deep etching was carried out on the longitudinal cut samples.

For the carbon-free alloys, the electron backscatter diffraction (EBSD) method was used at a magnification of 250× to examine the single-crystal crystallographic orientations with respect to the misoriented regions. If any were present, they were examined in more detail at higher resolutions. To study the nucleation behavior of eutectics on carbides, the transverse samples were first etched for 5 s with a waterless Kalling reagent. Thus, micro-structure components with different crystal orientation than the single crystal were visible under the OM. The areas of interest were then marked with a circle by an installed diamond in the OM and polished for 2–3 h on the VibroMet. If etched surfaces were still present, marking and polishing were repeated until the marked samples were in the polished unetched state. The crystallographic orientation of these marked areas was investigated using EBSD measurements. The investigations of the crystal orientation were carried out on the transvers cut samples.

If the carbides were not visible in the cross-section, it was ground off step-by-step in order to confirm the nucleation mechanism of the eutectics on the carbides. This stepwise examination of the eutectics revealed their 3D morphology. By grinding the sample against the direction of growth or withdrawal, it was possible to determine where the eutectics nucleated and the nucleation mechanism on which it was based. In addition, EBSD measurements were also used for this investigation.

## 3. Results

### 3.1. Carbon-Free Alloys

In carbon-free alloys, micro-stray grains with a slightly different crystallographic orientation than the single crystal were formed in the interdendritic region during single-crystal solidification (Figure 2a,c). As illustrated in Table 4, they were only formed in the alloy CMSX-4. Since the lattice structure of the micro-stray grains was observed to be identical to that of the eutectics, they represented a preferred nucleating surface for the eutectics. The difference of the micro stray-grain compared with the eutectic was the morphology, as seen in Figure 2a,c,e,g.

In the carbon-free alloys, misoriented eutectics were also found in the interdendritic region due to homogeneous nucleation from the residual melt in the interdendritic regions (Figure 2b,d), which was observed in both CMSX-4 and CMSX-6. The phenomenon in the alloy CMSX-4, however, only occurred at high-temperature gradients of 3.3 K/mm and high withdrawal rates starting from 4 mm/min, as listed in Table 5. In CMSX-6, homogeneous eutectics were detected at all process parameters applied in this study.

In the Bridgman–Stockbarger process, it was noticeable that no homogeneous eutectics were formed in the alloy CMSX-4. In the Bridgman process, on the other hand, this effect was only seen up to two times per sample cross-section for the alloy CMSX-4, whereas plenty was contained in the CMSX-6 alloy. All in all, the eutectics were preferentially formed through homogeneous nucleation from the residual melt at higher cooling rates and in the CMSX-6 rather than in the CMSX-4 alloy.

### 3.2. Carbon-Containing Alloys

In carbon-containing nickel-based superalloys, carbides with different morphologies were formed in the interdendritic regions. As demonstrated in Figure 3 for alloys CMSX-6-LC1 and CMSX-6-LC2, the carbide core exhibited an octahedral shape, whereby the arms extended from the corners of the octahedra. The influence of withdrawal rates on the formation affinity of the arms can also be seen in Figure 3. For example, in the Bridgman–Stockbarger process, the arms appeared in the alloys only from a withdrawal rate of ≥1 mm/min. The ends of the arms were arrowhead-shaped (Figure 3e). As shown in Figure 3c,f, plates could also form between the arms at higher withdrawal rates. Furthermore, it was visible that the size of the octahedra decreased with increasing withdrawal rates and the branching of arms formed more complex structures at the withdrawal rate of 4 mm/min. At the withdrawal rate of 0.25 mm/min, no arms were formed; instead, several layers were formed around the octahedra (Figure 3a,d). Analogously, these formation mechanisms were also observed in the Bridgman experiments (Figure 4).

Similar carbide morphologies were also observed in the CM-247-LC and MAR-M-247 alloys. However, the carbides grew into larger carbides due to the higher carbon content compared with the CMSX-6 series. Nevertheless, the origin consisted of an octahedral core having extended dendrite-like arms at the corners. Due to the higher carbon content, large branched morphologies were formed, which could grow together and form closed chambers due to the plate formation between the arms (Figure 3g). At higher temperature gradients, as in the Bridgman–Stockbarger process, the carbides became smaller and more compact than those in the industrial Bridgman system. Furthermore, the carbide size decreased with increasing withdrawal rates. The carbides were limited in size by the dendrites and adapted in shape to the dendrites at the end of solidification.

The formation of complex intensely branched carbide arms with plates (e.g., Figure 3g) resulted in a closed-chamber structure which could entrap the residual melt. The residual melt then solidified differently from the single crystal. However, the chambers may not have been completely closed, but open in one or more directions. In which direction a carbide structure was closed and open was random, since the carbide could move and rotate freely at the early stage of growth if there was enough interdendritic space. From these results, it could be concluded that, if the carbide structure was open in the direction opposite to the growth direction, the single crystal could produce a small orientation deviation while growing into the residual melt entrapped in the carbide (Figure 5a,e,j,o). Furthermore, the entrapped melt could nucleate at the carbide surface when the carbide was completely closed against the growth direction. As a result, an identical crystallographic orientation of the carbide and the solidified entrapped melt was formed (Figure 5f,k,p). The different enclosed chambers in a carbide could also have slightly different orientations, as illustrated in the upper part of Figure 5k. The carbide chamber with the micro-stray grain could also be open in the solidification direction, causing the misorientation to grow into the interdendritic region, which is shown in the EBSD measurement in Figure 5b,g,l,q. Moreover, eutectics could also nucleate at micro-stray grains that had been formed in a carbide and grown into the residual melt, as shown in Figure 5b,g,l,q. This was based on the very similar lattice defect between the γ-phase and the eutectics, identical to the homogeneously solidified micro-stray grain in the CMSX-4 alloy (Section 3.1).

The mechanism of entrapping the residual melt in carbides was not detected at low carbon content, as in the CMSX-6-LC1, and increased with the addition of carbon (Table 6). The temperature gradient also influenced this effect, i.e., in the Bridgman–Stockbarger process with higher temperature gradients, this effect only occurred at higher carbon contents compared with the Bridgman process with lower temperature gradients. Furthermore, it could also be seen that this effect occurred preferably at higher withdrawal rates, since it did not occur in CM-247-LC W1 but occurred in the same alloy at a higher withdrawal rate of W4.

Eutectics could also nucleate directly on carbides and consequently adopt the crystallographic orientation of the corresponding carbide. This effect was observed both on blocky carbides in the alloy CMSX-6-LC2 (Figure 5c,h,m,r) and on Chinese-script-shaped ones in MAR-M-247 (Figure 5d,i,n,s). Stepwise grinding ensured that these eutectics were not nucleated on micro-stray grains as described previously.

The conditions in which eutectic nucleation was detected on a carbide are listed in Table 7. In the Bridgman process, it was shown that no nucleation of eutectics on carbides was detected at low carbon contents, such as CMSX-6-LC1, or at very high withdrawal rates, such as 5.5 mm/min. This phenomenon was also seen in the Bridgman–Stockbarger process with higher temperature gradients. However, the nucleation shifted to lower withdrawal rates and higher carbon contents. Consequently, in the Bridgman–Stockbarger process, the nucleation of eutectics on carbides was also no longer possible for the alloy CMSX-6-LC2 above a withdrawal rate of 1 mm/min. This nucleation mechanism was also observed in MAR-M-247 only up to Q-W1.

The number of eutectics nucleated on carbides per area was analyzed using mosaic micrographs over the entire sample cross-section of CMSX-6-LC2 and MAR-M-247 at Q-W0.25. In CMSX-6-LC2 Q-W0.25, one eutectic nucleated on a carbide per six mm^2^ was detected, while five eutectics were detected in MAR-M-247 Q-W0.25 in the same area. In addition, the sample MAR-M-247 Q-W1 was investigated and three misoriented eutectics were found in the same area (three misoriented eutectics per six mm^2^). In summary, the number of eutectics nucleated on carbides per area increased with increasing carbon contents and decreased with increasing withdrawal rates.

## 4. Discussion

### 4.1. Carbon-Free Alloys

The interdendritic orientation defect is related to the critical undercooling ability of the melt. The undercooling of CMSX-4 is 9 K and CMSX-6 is 50.4 K [11,19]. According to the literature, the increase of Re causes a decrease of critical undercooling of the material [20]. Since the CMSX-4 alloy has a lower undercooling capability than the CMSX-6 alloy, homogeneous nucleation of the γ-phase occurs due to the segregation effects in the interdendritic regions [20,21]. Consequently, the γ′-precipitates form from it. However, this defect occurs only in the Bridgman experiments due to the lower temperature gradient (or cooling rate), leading to a larger primary dendrite arm spacing (PDAS) [22,23]. Thus, a larger interdendritic volume is formed between the dendrites and, because of the undercooling, the micro-stray grains are formed out of the residual interdendritic liquid before the γ/γ′-eutectics. In summary, alloys with low critical undercooling capabilities and larger PDAS form misoriented micro-stray grains from the residual melt due to homogeneous nucleation.

The γ/γ′-eutectics usually nucleate on the γ-dendrites due to the nearly identical lattice structure. Thus, the eutectics have the same crystallographic orientation as the single crystal. The homogeneous nucleation of eutectics in the interdendritic regions results in the formation of randomly oriented crystals that deviate from the single crystal [17]. The preferred parameters for this mechanism are high withdrawal rates and/or high temperature gradients, since, under these conditions, elemental diffusivity decreases due to shorter solidification times. Consequently, the accumulation of γ-forming elements in the residual melt increases and constitutional undercooling is promoted. As a result, the conditions for homogeneous formation of eutectics are better at higher withdrawal rates [11,17]. The higher content of homogeneously nucleated eutectics in the alloy CMSX-6 is due to the higher PDAS compared with the CMSX-4 (Figure A1). This leads to a larger interdendritic area, which results in a larger distance to the dendrite and higher elemental enrichment, as well as undercooling. Thus, the probability of forming homogeneously nucleated eutectics increases due to the coarser dendrite structure and the higher segregation of the elements in the alloy CMSX-6.

### 4.2. Carbon-Containing Alloys

#### 4.2.1. Carbide Morphology

In all alloys used in this study, octahedra are formed at the beginning of carbide formation. This is the equilibrium shape due to the minimum interface energy between carbide and liquid [24,25]. Octahedra are mainly observed at low withdrawal rates and cooling rates [24]. Under equilibrium conditions, the growth of carbides consists of layer growth, which is illustrated in Figure 3d. This mechanism has already been described by Baldan et al. [26], Chen et al. [24], and Liu et al. [27].

With increasing carbon contents from CMSX-6-LC1, CMSX-6-LC2 to CM-247-LC, and MAR-M-247, the carbide morphology acquired a more complex structure. It changed from blocky and acicular to Chinese-script shape with increasing carbon contents and/or increasing cooling rates. These results supported the studies in the literature [4,28]. At low carbon contents, the carbide volume is small, so the blocky shape provides the smallest surface area [7,29]. Higher carbon contents lead to carbides that develop arms from the octahedral tips. The growth direction of the arms is perpendicular to each other [26,30]. Further increases in carbon contents result in the formation of carbides with cubic dendritic shapes [30,31] ending up in the formation of even more arms, which grow together with plates and form arrow-shaped tips, also described by Yu et al. [4].

As confirmed by Li et al., higher carbon contents lead to greater supersaturation in the later stages of solidification, which promotes the growth of well-developed secondary and tertiary dendrite arms [28]. Although Chinese-script-shaped carbides have a large surface area, a certain orientation relationship between the two phases provides a lower interfacial and deformation energy. Therefore, a solidification system with a large carbide volume has the lowest free energy when the shape of the carbides is Chinese-script [29].

The impact of the cooling rate, i.e., the thermal gradient and the withdrawal or growth rate, on carbide size can be clearly seen in Figure 3a–f. High cooling rates decrease the elemental diffusivity, resulting in more carbide-forming elements accumulating in the interdendritic regions. Consequently, higher undercooling occurs, leading to a smaller critical nucleation radius and thus easier nucleation and growth of carbides [32,33,34]. Hence, carbides are continuously refined with a further increase in cooling rate [7,26,31].

#### 4.2.2. Nucleation in Carbide Chambers

In this study, micro-stray grains were formed in Chinese-script-shaped carbides with a closed chamber, which was caused by heterogeneous nucleation on the carbides. This mechanism is possible since the carbides, as well as the γ-phase, have a face-centered cubic crystal structure [30,35]. However, these chambers can also be open and the direction of the opening is random, as carbides can move freely in the melt. Once the chamber is open in the growth direction, the micro-stray grain can grow into the residual melt (Figure 5b,g,l,q). Due to the same lattice structure, eutectics can also nucleate on these structures and adopt the crystallographic orientation.

The presence of this misoriented defect increased with higher carbon contents, as listed in Table 4. This phenomenon was not observed at low carbon contents, i.e., in the alloy CMSX-6-LC1. Furthermore, higher carbon contents were required for it to occur in the Bridgman–Stockbarger process than at lower temperature gradients in the Bridgman process. This is because carbide chambers were more likely to form at higher carbon contents and lower temperature gradients. The withdrawal rate also influenced this defect. An increase in withdrawal rates promoted the formation of closed chambers and more carbide branches were formed at higher withdrawal rates. All in all, there was a clear correlation between the micro-stray grain in the carbides to the carbide morphologies and thus to the carbon content, as well as the process parameters.

#### 4.2.3. Nucleation on Carbides

Eutectics can nucleate directly on carbides and adopt their crystallographic orientation. Since carbides are formed freely in the melt and thereby have a random crystallographic orientation, the eutectics also adopt this orientation and are independent of the orientation of the single crystal [17,36]. The fact that the eutectics nucleate on the carbides and not vice versa is explained by the semi-coherent interface, which always shows a dislocation on the eutectic side and not on the carbide side [16].

Based on these results, it could be concluded that the alloy composition and carbide composition did not affect the nucleation of eutectics on carbides in the alloys used in this study, since it was observed in all carbon-containing alloys, except for CMSX-6-LC1. Therefore, a direct correlation was solely attributed to the carbon content, or carbide morphology, since in CMSX-6-LC2, with the same base composition and only higher carbon content, the nucleation of eutectics on carbides was found. These results were new, compared with the literature. Additionally, a positive link between increasing carbon content and carbide content, size, and arms, as well as branching, was identified in this study.

During carbide formation, the carbide-forming elements, i.e., C, Ti, Ta, and Hf, were integrated from the residual melt. At the same time, non-carbide-forming elements accumulated around the carbide; these included elements such as Al, Co, Cr, and Ni, which accumulated around the crystallized carbides, as shown in Figure 6a. It then formed an environment that was ideal for the nucleation of γ/γ′-eutectics, since Co and Cr were members of the γ-forming elements and Al was a γ′-forming element, while Ni was incorporated in both. Despite the different lattice parameters requiring a larger nucleation energy, nucleation of the eutectics on the carbides could occur. The high energy due to the semi-coherent interface measured by Wang et al. [16] was overcome by elemental segregation around the carbides. Reaching a saturation level around the carbides led to significant undercooling and consequently facilitated eutectic nucleation. During growth, they exchanged their respective forming elements, resulting in a common growth mode (Figure 6b). This hypothesis of Wang et al. [16,17] was supported by the results of this study. For small carbides, as in the case of the alloy CMSX-6-LC1, insufficient eutectic-forming elements accumulated around the carbides in order to achieve this effect due to the small size of the carbides. This influence of the carbide size on the nucleation effect was investigated for the first time in this study. This nucleation effect of eutectics on carbides confirmed the eutectic reaction, since a peritectic reaction on the carbide was not possible. Furthermore, these results confirmed that the fine structure of the eutectics was formed first and the coarser γ′ was formed later. The larger the carbides, the more elements accumulated around the intermetallic phase and the higher the probability that an eutectic would nucleate.

In the alloy MAR-M-247, more misoriented eutectics on carbides per area were formed compared with those in CMSX-6-LC2 with the same process parameters, since, on the one hand, more carbides and, on the other hand, larger carbides were present due to the higher carbon content. This led to a higher accumulation of elements around the carbide and, consequently, more eutectics nucleated on carbides; this was investigated for the first time in this study and confirmed the “segregation of elements around the carbides” hypothesis of Wang et al. [16,17].

When increasing the withdrawal rate in the Bridgman–Stockbarger process, this phenomenon became less pronounced, since less and smaller carbides were present. Low withdrawal rates resulted in a more compact carbide morphology that had a good surface-to-volume ratio (Table 5). As reported by Wang et al., increasing the withdrawal rate leads to thinner carbides, resulting in less eutectic-forming elements being precipitated and accumulating around the carbides [17]. Consequently, saturation around the carbides was not reached and no eutectics nucleated on the carbides (Table 5, Bridgman–Stockbarger results). Another effect of an increased cooling rate is the reduction of the dendrite arm spacing [23], whereby the carbides become smaller due to the smaller interdendritic space. In addition, more eutectic-forming elements accumulate around the dendrites due to the shorter time for diffusion. As a result, the undercooling around the dendrites becomes larger, causing the eutectics to nucleate on the dendrites due to the nearly equal lattice parameter [37]. Due to the carbide and dendrite morphology, the probability of the eutectics nucleating on the carbides decreased when the cooling rate was increased (Table 5, W5.5).

When the temperature gradient was increased, the eutectics nucleated at lower withdrawal rates on the carbides, which was evident from the Bridgman–Stockbarger experiments, with higher temperature gradients compared with those in the Bridgman experiments. Again, this mechanism occurred due to the carbide morphology, which formed a finer structure and consequently had a low surface-to-volume ratio. As a result, less eutectic-forming elements accumulated around the carbide and nucleation was not possible due to the different lattice parameters. Furthermore, the dendrite arm spacing decreased with increasing temperature gradients [38], leading to smaller carbides and lower enrichment of elements in the interdendritic regions due to smaller diffusion distances. In summary, the nucleation mechanism of eutectics on carbides depended mainly on the carbide morphology and, in this context, on the carbon content and the process parameters. The influence of these parameters on the nucleation effect of eutectics on carbides was systematically investigated for the first time in this study.

## 5. Conclusions

During directional solidification, the following micro-structural components form in both carbon-free and carbon-containing nickel-based superalloys, which have a different orientation than the single crystal:

Carbon-free alloys:Micro-stray grains form in the interdendritic regions where a eutectic can also nucleate and thus assume a different crystallographic orientation compared with the single crystal. This occurrence of the misoriented micro-stray grain is dependent on the critical undercooling ability of the alloy.Eutectics can also nucleate homogeneously in the interdendritic residual melt. The preferred parameters for this mechanism are high withdrawal rates and/or high temperature gradients.

Carbon-containing alloys:
Chinese-script-shaped carbides are formed at high carbon contents or high cooling rates, forming closed chambers. In these chambers, micro-stray grains are also formed by nucleation on the carbide. When the carbide chamber is open in the growth direction, these misoriented micro-stray grains also grow into the interdendritic residual melt. Consequently, eutectics can also nucleate on the micro-stray grains and adopt their crystal orientation.Eutectics can nucleate directly on carbides. This mechanism is independent of the chemical composition of the carbides from the alloys used in this study. However, the mechanism only occurs with sufficient accumulation of eutectic-forming elements around the carbides, which depends on sufficient carbon content or carbide size and surface/volume ratio (morphology) of the carbide.

## Figures and Tables

**Figure 1 materials-16-04477-f001:**
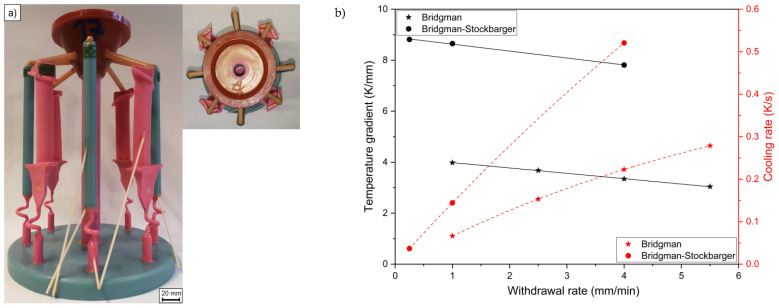
(**a**) Front view and top view of the wax cluster with the installed ceramic tubes for the thermocouples and (**b**) temperature gradient and cooling rate as a function of the withdrawal rate of the applied process parameters.

**Figure 2 materials-16-04477-f002:**
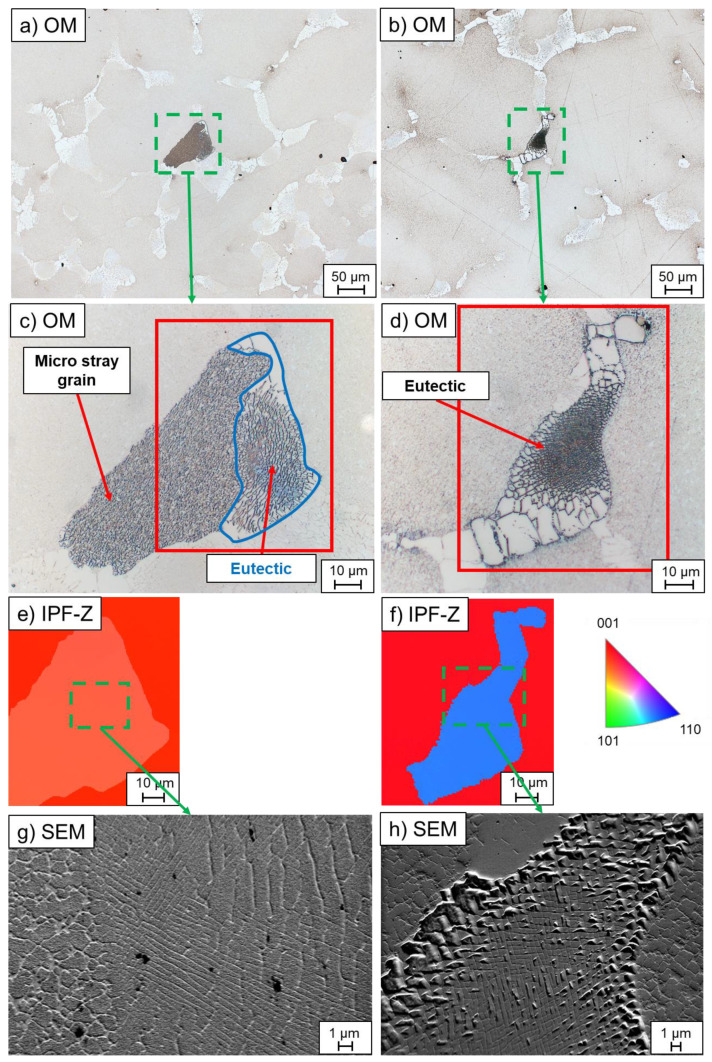
Optical micrographs and EBSD analyses on the z-orientation of homogeneously formed components with a deviating crystallographic orientation from the single crystal in carbon-free alloys: (**a**,**c**,**e**,**g**) micro-stray grain with nucleated eutectic (CMSX-4 W4); (**b**,**d**,**f**,**h**) homogeneously formed γ/γ′-eutectic (CMSX-4 W4) (OM—optical microscope; red rectangles represent the EBSD measurement area; IPF-Z—inverse pole figure in Z-/withdrawing direction).

**Figure 3 materials-16-04477-f003:**
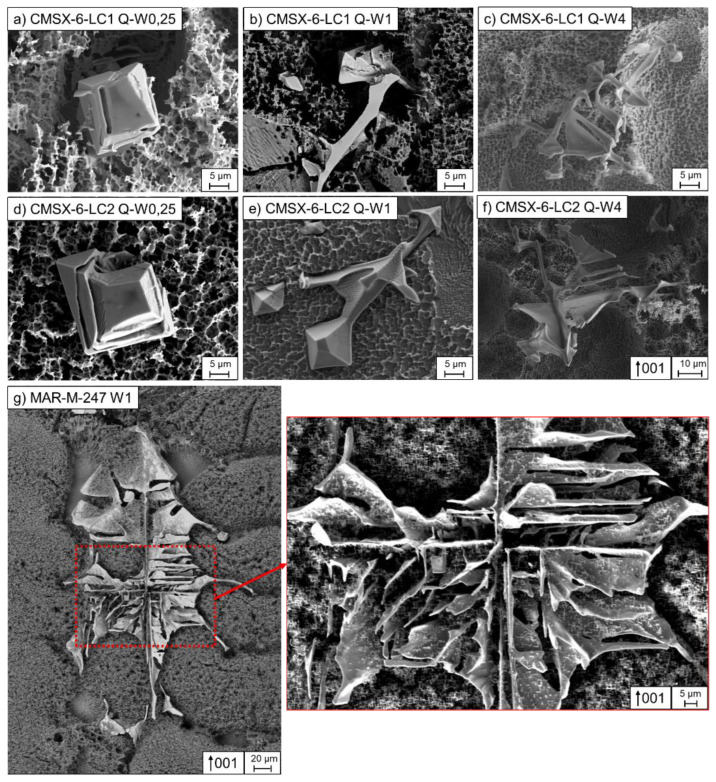
Carbide morphologies (etched with aqua regia): (**a**–**f**) octahedrons with outgrowing arms of the alloys CMSX-6-LC1 and CMSX-6-LC2 in the Bridgman process and (**g**) a complex Chinese-script-shaped carbide in MAR-M-247.

**Figure 4 materials-16-04477-f004:**
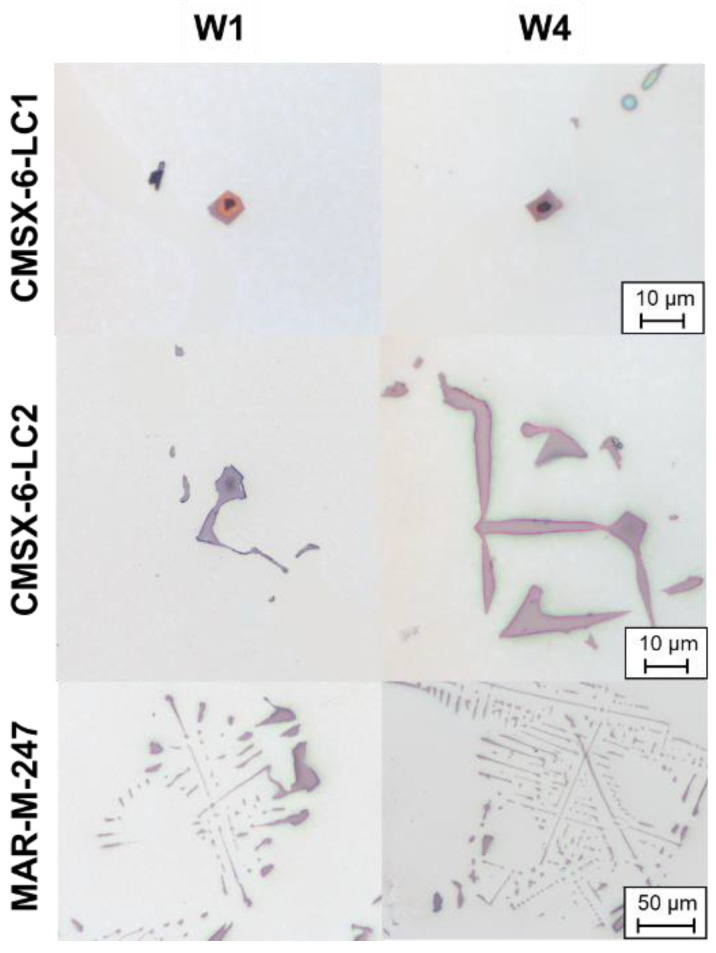
Carbide morphologies of the alloys CMSX-6-LC1, CMSX-6-LC2, and MAR-M-247 depending on the withdrawal rate.

**Figure 5 materials-16-04477-f005:**
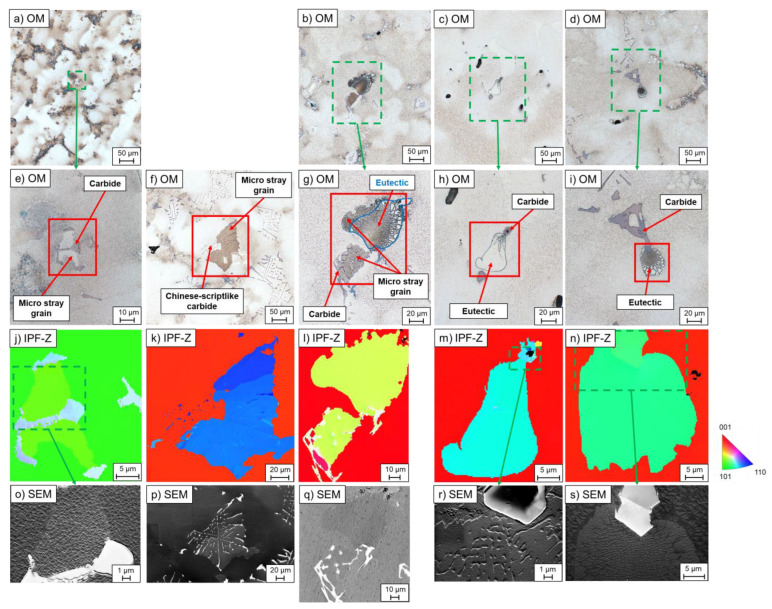
Optical micrographs and EBSD analyses on the z-orientation of the misoriented micro-structural components in carbon-containing alloys: (**a**,**e**,**j**,**o**) enclosed micro-stray grains similar to single crystal (MAR-M-247 Q-W4); (**f**,**k**,**p**) enclosed micro-stray grains with different chambers in carbide (MAR-M-247 W2.5); (**b**,**g**,**l**,**q**) eutectic nucleated on micro-stray grains (MAR-M-247 Q-W1); (**c**,**h**,**m**,**r**) eutectic nucleated on carbide in CMSX-6-LC2 (CMSX-6-LC2 W1); (**d**,**i**,**n**,**s**) eutectic nucleated on carbide in MAR-M-247 (MAR-M-247 W1) (OM—optical microscope; red rectangles represent EBSD measurement area; IPF-Z—inverse pole figure in Z-/withdrawing direction; SEM—scanning electron microscope).

**Figure 6 materials-16-04477-f006:**
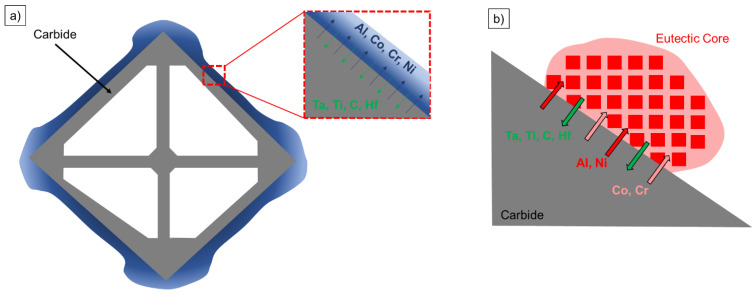
(**a**) Schematic illustration of the accumulation of eutectic-forming elements around a carbide cross section; (**b**) schematic illustration of the exchange of elements during eutectic and carbide growth.

**Table 1 materials-16-04477-t001:** Chemical composition of the applied nickel-based superalloys in wt.%.

Alloy	Cr	Co	Mo	W	Al	Ti	Ta	Re	Hf	C	B	Ni
CMSX-4	6.5	9.0	0.6	6.0	5.6	1.0	6.5	3.0	0.1	-		Bal.
CMSX-6	10.0	5.0	3.0	-	4.8	4.7	2.0	-	0.1	-		Bal.
CMSX-6-LC1	10.0	5.0	6.0	-	4.8	4.7	2.0	-	0.1	0.02		Bal.
CMSX-6-LC2	10.0	5.0	6.0	-	4.8	4.7	2.0	-	0.1	0.05		Bal.
CM-247-LC	8.1	9.2	0.5	9.5	5.6	0.7	3.2	-	1.4	0.07	0.015	Bal.
MAR-M-247	8.4	10.0	0.7	10.0	5.5	1.0	3.0	-	1.5	0.15	0.015	Bal.

**Table 2 materials-16-04477-t002:** Temperature gradient and cooling rates of all withdrawal rates.

	W1	W2.5	W4	W5.5	Q-W0.25	Q-W1	Q-W4
Temperature gradient	0.664	0.153	0.223	0.279	0.037	0.144	0.521
Cooling rate	3.98	3.68	3.42	3.04	8.82	8.65	7.81

**Table 3 materials-16-04477-t003:** Summary of experiments with all combinations of parameters and alloys that were carried out.

	W1	W2.5	W4	W5.5	Q-W0.25	Q-W1	Q-W4
CMSX-4	✓		✓		✓	✓	✓
CMSX-6	✓		✓		✓	✓	✓
CMSX-6-LC1	✓		✓		✓	✓	✓
CMSX-6-LC2	✓		✓		✓	✓	✓
CM-247-LC	✓		✓			✓	
MAR-M-247	✓	✓	✓	✓	✓	✓	✓

**Table 4 materials-16-04477-t004:** Incidence of micro-stray grains as a function of alloy, withdrawal rate and process (“✓” means detected; “x” means not detected).

	W1	W4
CMSX-4	✓	✓
CMSX-6	x	x

**Table 5 materials-16-04477-t005:** Incidence of homogeneous γ/γ′-eutectics as a function of alloy, withdrawal rate and process (“✓” means detected; “x” means not detected).

	W1	W4	Q-W0.25	Q-W1	Q-W4
CMSX-4	x	✓	x	x	x
CMSX-6	✓	✓	✓	✓	✓

**Table 6 materials-16-04477-t006:** Incidence of micro-stray grains enclosed in a carbide as a function of alloy, withdrawal rate and process (“✓” means detected; “x” means not detected).

	W1	W2.5	W4	W5.5	Q-W0.25	Q-W1	Q-W4
CMSX-6-LC1	x		x		x	x	x
CMSX-6-LC2	✓		✓		x	x	x
CM-247-LC	x		✓			x	
MAR-M-247	✓	✓	✓	✓	✓	✓	✓

**Table 7 materials-16-04477-t007:** Incidence of a eutectic nucleated on a carbide as a function of alloy, withdrawal rate and process (“✓” means detected; “x” means not detected).

	W1	W2.5	W4	W5.5	Q-W0.25	Q-W1	Q-W4
CMSX-6-LC1	x		x		x	x	x
CMSX-6-LC2	✓		✓		✓	x	x
CM-247-LC	✓		✓			✓	
MAR-M-247	✓	✓	✓	x	✓	✓	x

## Data Availability

Not applicable.

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
