# Peer review of "Influence of Process Parameter and Alloy Composition on Misoriented Eutectics in Single-Crystal Nickel-Based Superalloys"

_materials, 2023, doi:10.3390/ma16124477_

Round 1

Reviewer 1 Report

Review of the work: “Influence of Process Parameter and Alloy Composition on Misoriented Eutectics in Single-crystal Nickel-based Superalloys”

In this work, the authors seek to understand how processing conditions and chemical composition influence misoriented structures in the interdendritic spaces of single crystals of nickel-based superalloys.

The authors study the hypothesis of homogeneous nucleation of the eutectic in the interdendritic spaces of two carbonless nickel-based superalloys during directional solidification of single crystals under different processing conditions (withdrawal rate and cooling rate).

They also study the effects of different processing conditions on the morphology of carbides and their impact on the nucleation of disoriented eutectics in the interdendritic spacings of single crystals of four nickel-based superalloys with varying contents of carbon..

The experimentation uses an industrial Bridgman furnace and a Bridgman Stockbarger furnace. Metallographic specimens of longitudinal and cross sections of the single crystals are prepared, which are analyzed by optical microscopy and electron microscopy using EBSD.

It is an excellent work of impact in the aeronautical industry and of interest to Materials readers interested in producing turbine blades and advanced solidification processes.

I consider that it should be published in Materials after correcting the following minor corrections:

C1.-On page 1, line 42, it is mentioned that the g/ g´  eutectic is the last microconstituent to solidify from the remaining liquid of the nickel-based superalloys. In the results section, page 4,  paragraph including  lines 137-142, it is mentioned that, for CMSX-4 alloys, g/ g´ structures are formed in the interdendritic region with a slightly different orientation to that of the single crystal, and that, like the lattice structures of the misoriented g/ g´ structures are identical to those of eutectics, they represent a preferred surface for the nucleation of eutectics.

Include in this part of the text a paragraph with a brief explanation of the reasons why the misoriented structure g/ g´ is formed before the eutectic and what are the differences between both microconstituents.

C2.-In the discussion section, page 9, lines 265-267, the authors mention the differences in the eutectic nucleation habits in the interdendritic spaces shown by the CMSX-4 and CMSX-6 alloys because they present different undercooling capabilities.

Please include in the text a paragraph briefly explaining how the differences in the chemical elements' contents in Table 1 affect the undercooling potential for the CMSX-4 and CMSX-6 alloys.

Reviewer 2 Report

Dear authors, thank you for interesting articles.

I have some issues.

1) According to chapter 2, you have done a lot of experimental work. This article gives just a small part of it. You should choose, whether you want to give just qualitative results or quantitative as well.

Given results are qualitative only, OM+EBSD+indication of presence in yes/no manner, no results of phase fractions, particle sizes etc. But in discussion you are talking about correlations and trends, for which you are giving no data in the Results section.

2) I do not undestand what do you mean by "misoriented gamma/gamma prime structure. What is the difference to eutectics? In terms chemical composition, crystallography, occurence in solidification sequence e.g. Does if form from the liquid, or is it solid state precipiration? + any supporting data or literature data.

Please be carefully with using the word "misoriented". You are talking about misoriented phases or misoriented eutectics . It might be misleading for some readers. Could you call the specific phase "misoriented g/g' structure" by other words. This phrase could be understood as a decription of eutectics, too.

minor issues:

1) Introduction - images might be useful to illustrate the text, but not necessary

Line 35 - Dendritic solidification occurs not only during DS, also in polycrystal casting.

Table 1 - Did you check carbon content in carbon-free alloys?

Line 109 - How many metallographic specimens were prepared? Which of them did you use in your study, longitudinal or transversal? From blades or from cylindrical specimens (Fig 1a)?

Line 121 - How was the "marking" performed?

Figure 1b - This set of experiments was performed completely for each alloy?

chapter 2 - Were the castings analyzed in as-cast state, or was any Heat Treatment performed?

3 Results - Macroimages will be useful to illustrate the dendritic structure and position of described phenomenons (eutectics, misoriented g/g', carbides) in it. E.g. ss in literature nr. 19.

Line 148-150 - This sentence belongs probably to Discussion part?

Tables 2, 3, 4, 5 - Might be interesting to give this data in plot similar to Fig. 1b. Or maybe with coordinates cooling rate vs temperature gradient. At least in discussion section.

Line 160 - "In a quantitative comparison ..." this is 1 illustration of my main issue. You have given no quantitative date in this article to be able to make any quantitative comparison.

Figure A1 - Why is it in Appendix section?

Figure A2 - PDAS might be useful but you have given no single word about it in the article before, e.g. in Section 2

Figure 3 - Nice pictures. EDS, EBSD data of carbides will be nice. but I undestand it is not the point of this article.

Fig. 4a - OM insted of LiMi?

4.2 Discussion of C containing alloys - is long, please use subchapters

4.2 - is hard to follow, citing many results from the literature. what is new from your experiment? what is supporting or contradicting previous results?

Reviewer 3 Report

This manuscript, titled "Influence of Process Parameters and Alloy Composition on Misoriented Eutectics in Single-crystal Nickel-based Superalloys" by Tobias Wittenzellner et al., describes a study investigating the effects of varying cooling conditions and temperature gradients on both carbon-free and carbon-containing nickel-based superalloys. The samples were prepared using the Bridgman and Bridgman-Stockbarger methods, and the analysis involved the use of optical microscopy, scanning electron microscopy (SEM), and electron backscatter diffraction (EBSD) techniques. The paper examines the γ/γ'-structures and eutectics within dendrites and interdendritic regions, providing a comprehensive discussion on the nucleation and formation of these misoriented microstructure components. Overall, the manuscript is well-presented, particularly in the discussion section. However, there are several questions that need to be addressed before recommending acceptance.

First of all, The discussion is solid, and the conclusion is summarized very well. However, the description of the experimental setup, particularly the sample-to-data correspondence, needs improvement. For example, but not limited to:

·       In line 96-97, the sentence "some of the single-crystal cylindrical samples produced from..." could be enhanced by providing specific names for the samples instead of using vague terms like "some" or "part."

·       In line 104-105, the sentence "The overall process parameters of the two sets of experiments, together with the resulting cooling conditions, are summarized in Figure 1b" would benefit from an additional description of the figure. It is important to consider if this would affect the subsequent discussion, as the cooling rate and temperature gradient are interrelated.

·       Given that the temperature gradient is a crucial aspect of the discussion, providing specific values in a table would be helpful for readers' convenience. A supplementary table could be included for this purpose.

·       Figure 2 does not specify which sample or under what conditions the figure was obtained.

·       In line 255-258, there is some confusion regarding the units and numbers. Please clarify these details.

Secondly, Can the Bridgman method be compared with the Bridgman-Stockbarger method samples? Although both methods are based on similar principles, they are distinct methods utilizing different equipment. In this paper, the only discussed difference is the temperature gradient, disregarding the different crystallization processes associated with these two methods.

Thirdly, The abbreviations in the figures need clarification. For instance, in Figure 2, what does "IPF-Z" mean in Figure 2c and 2d? Additionally, in Figure 4a, what does "LiMi" stand for? Figures 4f, 4g, 4h, 4i, and 4j also include "IPF-Z" without explanation.

Round 2

Reviewer 2 Report

 Dear authors,

thank you for themodifications. I have some more questions.

Chapter 3.1

Micro stray grains - are not everywhere (text, figure 2 caption, table 4 caption etc)

Line 179 – No chemical compositions comparison is given

Line 180 – probably reference to Figure 2 a and b

To comparison of eutectics and micro stray grains – morphology is different, but the gamma prime particles in Figure 2a seem to me closer to eutectics than to ordinary grain gamma prime precipitates.

Is the misorientation of the micro stray grains small (as in Fig. 2a) in all cases, or can be found larger?

Macroimages will be useful also in case of carbon-free alloys.
